

**Measuring ecosystem nitrogen status: a comparison of proxies**
Maya Almaraz[1], Stephen Porder[1]
[1]Department of Ecology and Evolutionary Biology, Brown University, Providence, 02912, USA
*Correspondence to:* Maya Almaraz (maya_almaraz@brown.edu)
**Keywords:** nitrogen availability, nutrient limitation, $\delta^{15}N$, nitrogen mineralization, dissolved
organic nitrogen
**Abstract**. There are many proxies used to measure nitrogen (N) availability in watersheds, but
the degree to which they do (or do not correlate) within a watershed has not been systematically
addressed. We surveyed the literature for intact forest or grassland watersheds in which several
metrics of nitrogen availability have been measured. Our metrics included: foliar $\delta^{15}N$, soil $\delta^{15}N$,
net nitrification, net N mineralization, and the ratio of dissolved inorganic to organic nitrogen
(DIN:DON) in soil solution and streams. Not surprisingly, the strongest correlation (Kendall's
tau) was between net nitrification and N mineralization ($\tau=0.61$, $p<0.0001$). Net nitrification was
correlated with foliar and soil $\delta^{15}N$ ($p<0.05$), while net N mineralization was correlated with soil
$\delta^{15}N$ but not foliar $\delta^{15}N$. Foliar and soil $\delta^{15}N$ were correlated across tropical sites ($\tau=0.68$,
$p<0.0001$), but not in temperate sites ($\tau=0.02$, $p>0.05$). To our surprise, the only significant
correlation we found between terrestrial- and water-based metrics was that of net N
mineralization with stream DIN:DON ($\tau=0.62$, $p=0.004$). Given both soil $\delta^{15}N$ and stream
DIN:DON are used to infer long-term N status, their lack of correlation in watersheds merits
further investigation.



**1.0 Introduction**

Nitrogen (N) limitation to primary production is widespread in both terrestrial and

aquatic ecosystems, and variation in N availability drives differences in ecosystem properties
across space and time (Vitousek and Howarth, 1991; Elser et al., 2007; LeBauer and Treseder,
2008). Nevertheless, quantifying N availability over timescales that are relevant in ecosystems is
non-trivial. Short timescale measurements of N availability in soil are common (e.g. inorganic N
pools, N mineralization and nitrification; Binkley and Hart, 1989; Sparks et al., 1996), but such
short-term proxies are influenced by both short and long-term drivers, and thus it is difficult to
know whether such short-term proxies can be used to infer N status over long timescales. For
example, measured net mineralization and nitrification in arctic tundra is commonly less than
annual plant uptake (Schimel et al., 1996; Schmidt et al., 1999), and annual N budgets based on
short-term measurements are difficult to balance (e.g. Magill et al., 1997). While N status over
longer temporal, and larger spatial, scales are relevant to many ecosystem properties and their
response to global change, it is more difficult to measure.

Land-based investigations of N cycling commonly measure extractable N, N

mineralization, and nitrification, which give a snapshot of N status over minutes to days (Binkley
and Hart, 1989; Robertson et al., 1999). Some researchers also use lysimeters to quantify
dissolved N losses from below the rooting zone (Hedin et al., 2003; McDowell et al., 2004;
Lohse and Matson, 2005) on a similar timescale. Repeated measurements give longer timescale
information, but even the longest studies are short relative to ecosystem development.

In addition to these short-term proxies, there are two relatively common measurements

that are thought to average over space and/or time. The first is the ratio of dissolved inorganic
(DIN) to organic (DON) N concentration lost from ecosystems in solution. Losses of DIN are



considered controllable by biota, and thus should be low if soil N is in short supply. In contrast,
most DON is not accessible to plants, and thus represents a loss beyond biotic control (Hedin et
al., 1995; Figure 1). Thus low DIN:DON in streams has been used to infer relative N-poverty in
watersheds (e.g. McDowell and Asbury, 1994; Perakis and Hedin, 2002; Brookshire et al., 2012).
The few sites where such measurements have been made over decades (e.g. the Luquillo
Mountains of Puerto Rico, Harvard Forest in Massachusetts, Hubbard Brook LTER in New
Hampshire; McDowell et al., 1992, McDowell et al., 2004, Bormann and Likens 2012) suggest
stream DIN:DON is not particularly variable over this timescale, and thus this metric may
integrate over time as well as space (W.C. McDowell, pers. comm.). It is common that
researchers using DIN:DON to infer ecosystem N status implicitly assume that a few
measurements are indicative of longer-term patterns (e.g. Perakis and Hedin, 2002; Brookshire et
al,. 2012).

In contrast to stream DIN:DON, soil $\delta^{15}$N integrates solely over time, and at steady state

reflects the isotopic signature associated inputs (N fixation and/or deposition) and fractionation
associated with outputs (Handley and Raven, 1992). The major N loss pathways (denitrification,
and to a lesser extent nitrate leaching) discriminate against $^{15}$N, which thus remains in relative
abundance in N-rich soils (Hogburg 1997; Martinelli et al., 1999; Craine et al., 2009; Houlton
and Bai, 2009, Craine et al., 2015; Figure 1). To some degree foliar $\delta^{15}$N reflects soil $\delta^{15}$N
(Amundson et al., 2003), but there can be fractionation between bulk and soil solution N pools
(Hogburg, 1997), as well as during N uptake by roots and mychorrhizae (Hobbie et al., 2009).
For this reason, foliar $\delta^{15}$N may display greater variability between species in a single site than
the bulk soil $\delta^{15}$N (Vitousek et al., 1989; Nadlehoffer et al., 1996).



Given that these proxies for N availability function over different spatial and temporal
scales, we asked which were correlated in watersheds where several measurements have been
made in the same place and at roughly the same time. We were particularly interested in whether
short-timescale measurements (nitrification, mineralization) correlated with the more temporally
(foliar and soil $\delta^{15}$N) and spatially (stream DIN:DON) integrated proxies. Unlike previous
reviews (Sudduth et al., 2013) we focus solely on unmanaged systems where we were able to
compare plant, soil, soil solution and stream proxies.

**2.0 Methods**

**2.1 Literature Review**
We surveyed the literature and contacted individual investigators to gather data from
forested and grassland watersheds where more than one proxy of long-term N availability had
been measured. We focused on the most commonly-used proxies for N status: foliar (n=78) and
surface soil $\delta^{15}$N (n=104; <20 cm depth), net nitrification (n=86; <20 cm depth), net N
mineralization (n=88; <20 cm depth), the ratio of dissolved inorganic to organic N forms
(DIN:DON) in soil solution below the rooting zone (n=43; >20 cm depth), and stream DIN:DON
(n=32). We chose these metrics because other authors have suggested that they are indicative of
soil nutrient status (Martinelli et al., 1999, Amunson et al., 2001, Brookshire et al., 2012; Figure
1), and because they thought to integrate N fluxes on different timescales (e.g. soil $\delta^{15}$N
integrates N losses over decades while net N mineralization rates integrate inorganic N
production over days; Binkley and Hart, 1989, Hogburg 1997). Soil values were from the
mineral soil only, and were preferentially collected in the 0-10 cm range, however if soil samples



were not in 10 cm increments, we selected the increment that was most similar (e.g. A horizon,
0-5 cm, 0-15 cm), and no deeper than 20 cm.
We used the search engines Web of Science and Google Scholar and searched key words:
"nitrogen", "15N", "natural abundance", "mineralization", and "dissolved organic nitrogen",
"*watershed name*". References in papers that resulted from the keyword search were then used to
gather additional data. We limited our search criteria to studies that took place in intact forest or
grassland ecosystems.
We collected data from 141 watersheds across a broad climatic range (Figure 2), in which
at least two of the six N proxies of interest had been measured. We used DataThief II software
(version 1.2.1) to extract data from figures when data were not available in text or tables. When
necessary, data were converted to standardize units.
From each paper we collected the following site description data: country, site, watershed,
biome, ecosystem type, latitude, longitude, elevation (m), mean annual temperature (MAT; °C),
mean annual precipitation (MAP; mm yr$^{-1}$), N deposition rate (kg N ha$^{-1}$ yr$^{-1}$), soil depth (cm),
soil solution (lysimeter) depth (cm), and N mineralization method. In order to control for
methodological differences, we limited our net nitrification and N mineralization methods to
those which used lab or buried-bag incubations (Boone, 1992; Piccolo et al., 1994), and
eliminated methods such as ion resin exchange beads or $^{15}$N tracer techniques (Binkley et al.,
1986; Hart and Firestone, 1989; Davidson et al., 1991; Templer et al., 2008). Site description
data were gathered from other sources when they were not in the original publication.
When data were missing, or we were uncertain about location or collection method, we
contacted the authors to request unpublished data, elucidation of data collection, data reduction,
or soil samples. For five watersheds (Puerto Rico's Pared, Sonadora, Bisley, Tronoja watersheds



and Hubbard Brook's watershed 6) we collected soil that we analyzed for $\delta^{15}$N. In Puerto Rico,
we collected mineral soil samples (0-10 cm) in replicates of five using an open side soil sampler
from locations that were >3 m away from the stream. Replicate samples were combined in a
Ziploc bag, air-dried and shipped to the Marine Biological Laboratory for analysis. Colleagues at
Hubbard Brook collected three replicate horizon B samples for us from several soil pits dug
across an elevation gradient in watershed 6 (Christopher Neill, personal communication), which
were air-dried at the Marine Biological Laboratory prior to analysis.

**2.2 Soil Sample Analysis**
The few soils we analyzed in house for $\delta^{15}$N were homogenized, sieved (2 mm) and
ground using a mortar and pestle. We analyzed samples at the Marine Biological Laboratory
Ecosystem Center Stable Isotope Laboratory for $\delta^{15}$N using a Europa 20-20 continuous-flow
isotope ratio mass spectrometer interfaced with a Europa ANCA-SL elemental analyzer. The
analytical precision based on replicate analyses of $\delta^{15}$N of isotopically homogeneous
international standards was ± 0.1 ‰.

**2.3 Statistics**
Five of our six variables were not normally distributed, so we used a non-parametric
Kendall tau rank test in R (version 2.11.1), to determine the significance of correlations.
Kendall's tau evaluates the degree of similarity between two sets of ranked data and generates a
smaller co-efficient as the number of discordant pairs between two ranking lists becomes greater
(Abdi 2007). The Kendall tau rank test is well suited for these comparisons as it is not sensitive
to missing data and outliers, it measures both linear and non-linear correlations, and generates a





more accurate p-value with small sample sizes (Helsel and Hirsch, 1992; Raike et al., 2003). We
corrected for multiple comparisons by reporting Bonferroni adjusted p-values for each of our 15
comparisons (Bland and Altman, 1995). We removed a single stream DIN:DON value from
Cascade Head, Oregon, as it was ~20 times higher than the mean of all other stream values
(Compton et al., 2003); however removing this outlier had little effect on the correlations.

**3.0 Results**
Most terrestrial-based proxies that integrate across long and short timescales were
significantly correlated. Soil $\delta^{15}$N was positively correlated with both net nitrification (n=58,
$\tau$=0.39, $p$=0.0002) and N mineralization (n=58, $\tau$=0.38, $p$=0.0005). Foliar $\delta^{15}$N was also
positively correlated with net nitrification (n=41, $\tau$=0.46, $p$=0.0003), but not with N
mineralization (n=40, $\tau$=0.25, $p$>0.05; Figure 2).
Not surprisingly, we found significant correlations between terrestrial-based proxies that
measure nutrient availability on similar timescales. Foliar $\delta^{15}$N was positively correlated with
soil $\delta^{15}$N (n=67, $\tau$=0.34, $p$=0.0006). There was also a positive correlation between net
nitrification and N mineralization (n=84, $\tau$=0.61, $p$<0.0001; Figure 3).
Despite the correlation between most terrestrial-based measurements of N availability,
terrestrial metrics did not exhibit similarly robust relationships with that of water-based proxies.
No metric was significantly correlated with soil solution DIN:DON (n=43, $p$>0.05), and net N
mineralization was the only metric to correlate with stream DIN:DON (n=17, $\tau$=0.62, $p$=0.004).
Soil solution and stream DIN:DON data were not correlated (Figure 3). All of the data in Figure
3, and their original sources, are available in Supplemental Table 1.



The lack of relationship between water-based and terrestrial-based metrics lead us to ask
questions about variability of soil solution and stream DIN:DON across environmental gradients.
We found that solution DIN:DON was not correlated with lysimeter depth (n=37, $p$>0.05).
Solution DIN:DON was positively correlated with temperature (n=43, $\tau$=0.22, $p$=0.04) and
negatively correlated with elevation (n=34, $\tau$=-0.25, $p$=0.04). To our surprise, solution
DIN:DON was higher in temperate than in tropical regions ($p$=0.02). Stream DIN:DON was not
correlated with elevation, temperature, precipitation or N deposition (n=32, $p$>0.05).
Some relationships between proxies differed with latitude. Soil $\delta^{15}$N correlated with
foliar $\delta^{15}$N (n=24, $\tau$=0.68, $p$<0.0001) in tropical, but not temperate, regions. Conversely, net N
mineralization was correlated with stream DIN:DON (n=10, $\tau$=0.78, $p$=0.03), and foliar $\delta^{15}$N
was correlated with net nitrification (n=21, $\tau$=0.49, $p$=0.04), in temperate but not tropical areas.
The only significant correlation across both tropical (n=26, $\tau$=0.53, $p$=0.003) and temperate
(n=54, $\tau$=0.62, $p$<0.0001) biomes was net nitrification with N mineralization.

**174    4.0 Discussion**

The metrics presented here are typically interpreted to fall into one of three categories: 1)
long-timescale (decades to centuries) integrators of soil N losses (foliar and soil $\delta^{15}$N; Martinelli
et al., 1999, Craine et al., 2015), 2) short-timescale direct measures of N transformations
(mineralization, nitrification; Vitousek et al., 1982), and 3) short-medium timescale (weeks to
years) measures of hydrologic N losses that are influenced by N availability in a catchment (soil
solution and stream DIN:DON; Hedin et al., 1995; Perakis and Hedin, 2001). Our data suggest
that correlations between category 1 and 2 metrics are robust, and that short-term soil assays may
capture similar patterns as inferred by long-term plant and soil-based proxies. However, the lack





of correlation between long-term terrestrial proxies (plant and soil $\delta^{15}N$) and both soil solution
and stream DIN:DON is interesting, as several authors have suggested that both types of proxies
give insight into ecosystem N status (Vitousek et al., 1982; Hedin et al., 1995; Martinelli et al.,
1999; Perakis and Hedin, 2001; Amundson et al., 2003; Brookshire et al., 2012).

It is particularly interesting that stream DIN:DON was not correlated with soil $\delta^{15}N$ as both

are proxies used to infer long-term N status. There is a wealth of literature that uses stream
DIN:DON to infer large spatial and temporal scale patterns in N availability (Hedin et al., 1995;
Perakis and Hedin, 2002; McDowell et al., 2004; Fang et al., 2008). Similarly, many studies
interpret soil $\delta^{15}N$ as an integrator of N losses over time (Martinelli et al., 1999; Houlton et al.,
2006; Houlton and Bai, 2009, Craine et al., 2015). These are the only two proxies for N status
that integrate over relatively long timescales, and their lack of correlation warrants more careful
consideration. We note that stream DIN:DON is sensitive to N deposition, and that relatively
pristine settings have a lower DIN:DON than polluted ones (Perakis and Hedin, 2001). But in
our dataset, N deposition was not correlated with stream DIN:DON ($\tau=0.03$, $p>0.05$), so it is
unlikely that N deposition is responsible for the lack of correlation between these two long-term
proxies.

Another surprise from our dataset is that soil solution DIN:DON was not significantly

correlated with any other metric, not even with stream DIN:DON, despite ~40% of papers in our
dataset reporting both soil solution and stream DIN:DON in the same watershed. While the
correlation between soil solution DIN:DON below the rooting zone and N availability has been
documented across gradients in soil age and fertility (Hedin et al., 1995), this correlation was not
found across the range of sites examined here. We found no correlation between soil solution
DIN:DON and lysimeter depth, suggesting that the majority of N transformations responsible for



the discontinuity between soil solution DIN:DON and that of terrestrial metrics are likely
occurring within the rooting zone. Soil solution DIN:DON was sensitive to environmental
variability – it decreased with increasing elevation and increased with temperature – whereas
stream DIN:DON was not, suggesting that further N processing below the rooting zone may also
occur to disconnect soil solution and stream N concentrations. From these data, at least, it does
not seem soil solution DIN:DON can be used infer terrestrial N status across this suite of
unmanaged sites.  These data also do not support the idea that soil solution DIN:DON is
representative of N forms that leach into streams (Binkley et al., 1992; Pregitzer et al., 2004;
Fang et al., 2008).

While nitrate ($NO_3^-$) removal along flow paths can reduce stream $NO_3^-$ (Vidon et al., 2010),

with higher removal in forested watersheds (Sudduth et al., 2013), DON has been shown to be
relatively resistant to removal pathways such as decomposition and biologic uptake (Carreiro et
al., 2000, Neff et al. 2003). We found no correlation between stream and soil solution DIN:DON,
suggesting that variability in $NO_3^-$ removal (relative to DON) along flowpaths below the rooting
zone of undisturbed ecosystems may explain this lack of correlation. The extent to which
riparian zones influence nutrients varies spatially with geomorphology, soil texture, vegetation,
and riparian zone development (McDowell et al., 1992, Mayer et al., 2007); and soils with high
rates of leaching to ground water may bypass riparian processing. As nutrients leach down the
soil profile, denitrification, biologic uptake, and storage are all potential mechanisms that could
alter soil solution and stream N species concentrations. Investigation of soil profile processes and
riparian zone spatial variability may help determine where and when watershed-scale N status
can be inferred from these proxies.



Although we found that temporal (soil $\delta^{15}N$) and spatial (stream DIN:DON) integrators of
watershed N were correlated with short-term proxies (net nitrification and net N mineralization),
stream DIN:DON did not correlate very well with most of the soil-based metrics of N
availability or each other. Explicit comparisons of these proxies to each other, with a focus on
how they are influenced by hot-spots, hot-moments, biological diversity, and N transformation
between the soil-stream interface, will enhance their utility for understanding N availability at
the ecosystem scale.

**236      5.0 Conclusions**

The labor and expense associated with fertilization studies to assess nutrient limitation
requires that we develop proxies to infer soil nutrient status. While nitrification and
mineralization most frequently correlated with other metrics, they are short-term proxies that
vary over short spatial and temporal scales. Soil $\delta^{15}N$ and dissolved N losses from streams are
long-term integrators of N loss that have been relied on to advance our understanding of N
cycling at the global scale (Martinelli et al., 1999; Amundson et al., 2003; Hedin et al., 2003;
Brookshire et al., 2012), however their lack of correlation brings to light a need to better
understand how these terrestrial and stream-based metrics vary in relation to each other and with
nutrient limitation.
Understanding ecosystem N status at the watershed and landscape scale is a first step towards
projecting their response to climate change and environmental pollution (Aber et al., 1998; Oren
et al., 2001; Reich et al., 2004). Soil N status can determine the rate at which detrimental N
losses occur, such as $NO_3^-$ (a drinking water contaminant) and nitrous oxide (a potent greenhouse
gas). Furthermore, it is becoming more evident that projections regarding the potential for a



terrestrial $CO_2$ sink, and concomitant feedbacks to the trajectory of climate change, are
dependent on the nutrient status of soils (Thornton et al., 2007; Zaehle et al., 2010; Wieder et al.,
2015). The health and environmental implications of soil N status heighten the need to develop
methodology to adequately assess long-term soil N availability.

**6.0 Author contribution**
M. Almaraz and S. Porder conceived research and designed study. M. Almaraz collected data
and performed statistical analyses. M. Almaraz and S. Porder wrote the manuscript.

**7.0 Acknowledgments**
We want to thank J. Campbell, C. Neill, and W. Wilcke for soil samples. M. Otter, C. Tamayo
and C. Silva for help with analyses. MA and SP received funding from NIH IMSD
R25GM083270, NSF DDIG GR5260021 and NSF EAR 1331841.





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



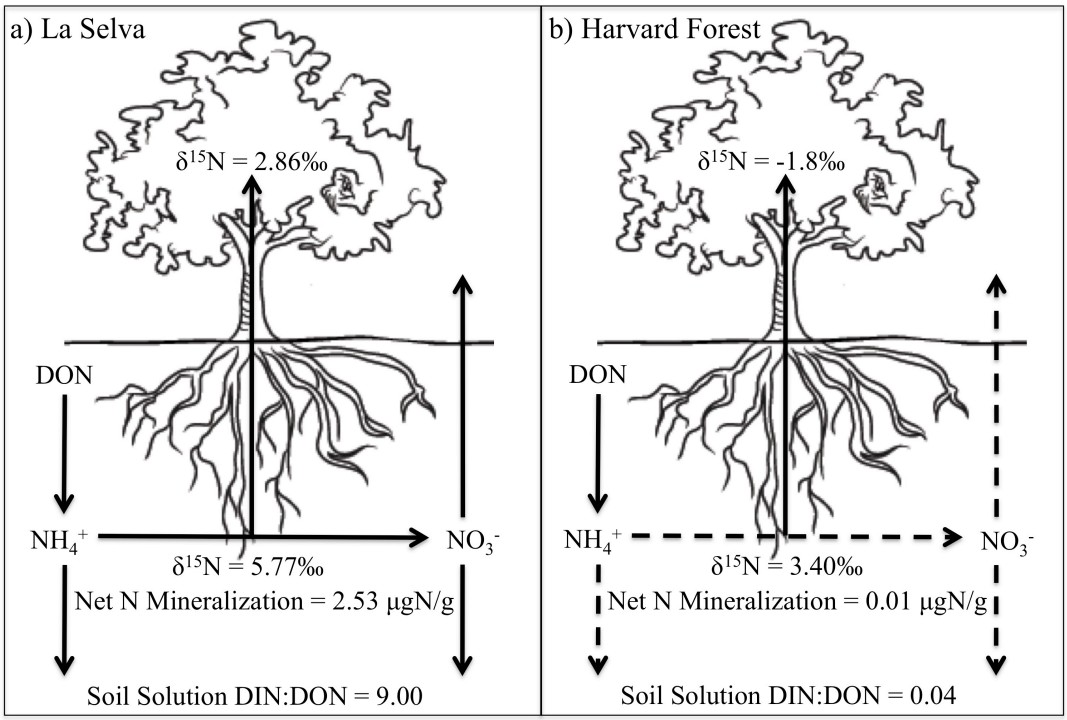

**Figure 1.** Nitrogen availability values for a) a nitrogen rich tropical forest at the La Selva field station in Costa Rica, and for b) a nitrogen limited temperate pine forest at Harvard Forest, Massachusetts. Solid and dotted lines represent the relative magnitude of fluxes (i.e. net N mineralization, denitrification to the atmosphere, dissolved organic and inorganic nitrogen leaching), which are contingent on ecosystem nitrogen status.






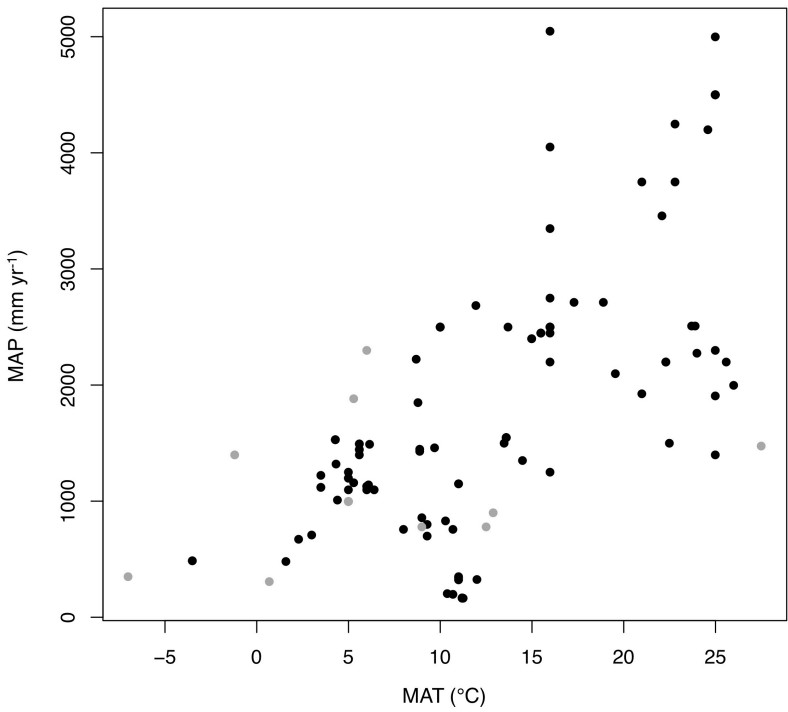


**Figure 2**. Distribution of grassland (grey) and forest (black) watershed mean annual temperature
(MAT; °C) and mean annual precipitation (MAP; mm yr$^{-1}$) included in meta-analysis.





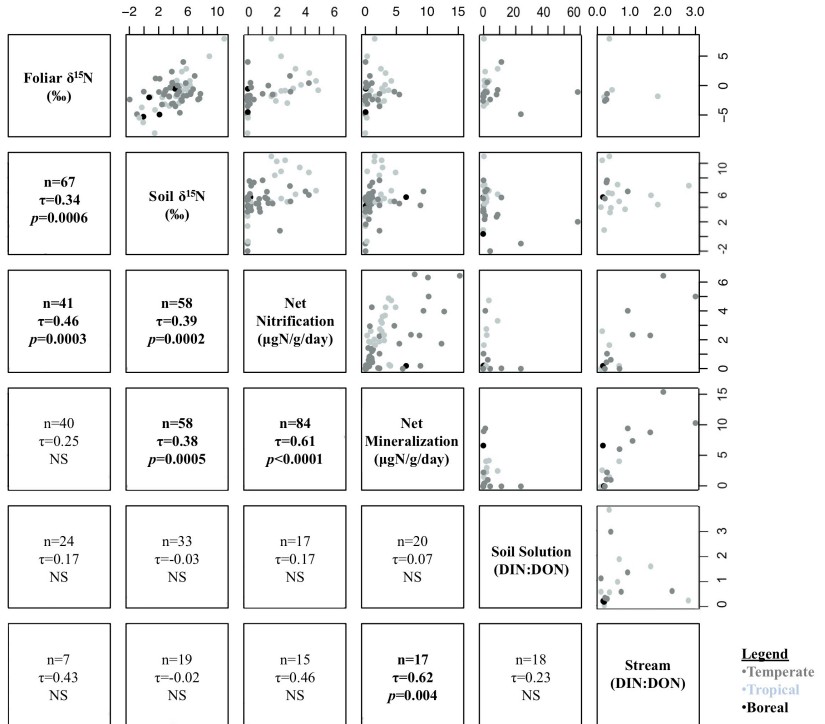

481

**Figure 3**. Correlation matrix of N status proxies (foliar and soil $\delta^{15}$N, net nitrification and N

mineralization (<20 cm), the ratio of dissolved inorganic to organic N forms (DIN:DON) in soil

solution below the rooting zone (>20 cm), and the DIN:DON in streams. Data are above the

diagonal, summary statistics are below. NS signifies correlations that were not significant

($p$>0.003).

487

488