# Peer review of "Measuring ecosystem nitrogen status: a comparison of proxies"

_Biogeosciences, 2016_

## Referee Comment (RC1) · Anonymous Referee #1 · 18 Apr 2016

This study presents a comparison of different proxies for assessing the nitrogen status of terrestrial ecosystems. Such proxies are essential if we are to assess the influence of nitrogen on these ecosystems, but various proxies exist and it is yet unclear which proxies are to preferred above others, and how the different proxies relate to one another. The current study addresses primarily the latter. Although I find this a useful effort, I think especially the analyses require revisions to improve the quality of this study.

General comments: As the study is now it is only indicative of the relationship between different proxies of nitrogen availability, and although differences between biomes are discussed, I think additional analyses are required to be able to draw firm conclusions regarding the difference among e.g. boreal, temperate and tropical systems. So, I suggest that the authors test statistically if for example the relationship between soil d15N

soil and foliar d15N differs between temperate and tropical systems (as suggested on l. 167-168).

Although I understand that the potential proxies are numerous and choices need to be made, I wonder why authors did not include soil C:N ratio. This ratio is often considered a good indicator of soil nutrient status (see e.g. Alberti et al 2015, iForest 8, 195-206). It is also an easy measurement to make so I advocate to include this ratio in the study.

I wonder if the authors could test for the influence of the distance between watershed and stream in the analysis of the correlation between soil proxies versus stream DIN:DON (perhaps with a subset for which this info is available).

Finally, I suggest that the authors make their dataset available. At minimum, a list of sites and coordinates should be given, but I hope authors can also provide the associated data for the various proxies.

Specific comments: l. 9: bracket after 'correlate' should come before 'correlate' l. 46: remove 'in solution' (dissolved is obviously in solution) l. 70: I suggest to rephrase 'we asked which were correlated' l.181-182: I disagree with this statement that category 1 and 2 proxies showed a robust relationship. In a dataset with huge variation (like this one), a significant correlation between 2 metrics does not necessarily imply a robust relationship, and visual inspection of the relationships in Fig. 3 does not support the statement. There is some correlation, but there is a lot of unexplained variance. l.194-198: How is d15N of soil (and plants) influenced by N deposition? If N deposition influences soil and plant d15N, the use of these proxies globally may not be ok because similarly rich sites would differ in d15N depending on the N source. l.211: 'that' after 'seem' and 'to' after 'used

---

## Referee Comment (RC2) · Anonymous Referee #2 · 28 Apr 2016

BG review This paper poses an interesting and important question about whether different metrics used to characterize N availability (which represent different spatial and temporal scales) are correlated. This topic is of potential interest to a broad group of researchers who consider N availability in their studies. The paper attempts to evaluate some underlying assumptions that are included implicitly or explicitly in interpreting ecosystem N dynamics. The scope of the analysis is not really clear from the paper. It is a little surprising that the authors did not include several large syntheses of similar data (Aber at al. 2003, BioScience, Pardo et al. 2006 Biogeochemistry, the CANIF study in Europe, Schulze 2000 Springer). The scope of the analysis is important, because it can be difficult to make assertions about different climatic zones or life forms unless enough variation is included among the samples to represent that observed

Several issues should be addressed:  ́c Nitrate leaching is referred to as if it were the driver of the fractionation that would lead to 15N-enrichment of material remaining in the ecosystem (soil, foliage). In fact the elevated nitrification which leads to an increase d15N of the plant available (including the nitrate that leaches to the stream) is the driver. The authors are, no doubt, well aware of this, but it is worth taking the trouble to be more precise for the reader less familiar with these dynamics. This should be addressed at several points in the paper. • The isotope literatures is not as current as it could be—I have given some examples of possible additional citations. • I assume that when the authors talk about long-term patterns and measures that are invariant temporally, that they mean in undisturbed systems. This should be stated explicitly, since over the long term, at many of these study sites, various disturbances have occurred which disrupt that N cycle and which would affect the values of these metrics. • Need to define what is meant by N status • More explanation about the differences between observed correlations in tropical versus temperate systems would be useful (why were foliar and soil d15N correlated in tropical, but not temperate?)

Abstract: 10 if space permit, include the region considered in this study is there a 'that' missing? i.e., given that both. . .

why 'Nevertheless"? what follows doesn't not seem to contrast with what was said in the first sentence.

don't really need 'such' on this line

I would suggest adding 'rates' after mineralization and nitrification, to make the comparison to another flux clearer. Also, the verb needs to agree with the subject is→are

33-4 This is an important point (basing annual budgets on short-term measures) and one that is often ignored.

34-6 There seems to be a word missing or a punctuation problem. Is 'are relevant' associated with scales or N status?

when is it not true that DON is not accessible to plants? Are those situations relevant here?

53-4 Presumably, the assumption is that DIN:DON integrates well over time in the absence of disturbance of any sort that disrupts the N cycle.

soil d15N 'represents the cumulative loss of N' is a bit narrow as a definition—more broadly, it reflects the net of all biotic transformations (that are fractionating)

this text needs to be changed to be more precise. Nitrate leaching is not a fractionating process. I presume the authors are using 'nitrate leaching' as shorthand to represent nitrification followed by nitrate leaching. It is worthwhile being clear about this for readers less familiar with isotope dynamics.

the phrase 'remains in relative abundance' is an odd locution. There are many standard ways to express this: d15N increased, the soil becomes enriched in 15N, etc.

insert comma after degree 62-3 see also Pardo et al 2006 for relationships between foliar and soil d15N

word missing after bulk→bulk soil

Additional more recent refs include: Cheng et al 2010 Plant and Soil 337:285-297; Pardo et al 2013 Biogeochemistry ; Templer et al. 2007 Oecologia, etc.

nitrification and mineralization measurements are not all short term. Buried bag measurements are often made over a year or longer. If the authors are referring only to short term lab incubations, they should call it net nitrification potential.

is foliar 15N temporally integrated?

80-82 do these numbers represent the number of sites included or do they include repeated measures at the same site? There should be some indication in the methods about the region and geographic extent of the sites included in the study. At the very least, mention should be made of the supplemental material.

87-90 Is this level of detail necessary?

What is meant by 'intact'? does this mean 'not fragmented'? Or is it intended to include disturbance as well? And if so, only anthropogenic disturbance (e.g., harvesting) or also natural (fire, wind, ice or pest events, etc.)?

Is there a list of the sites in supplemental information? (cite supplemental material here)

Is it appropriate to lump net nitrification potential measures with measures of nitrification? This should be justified.

105-6 This level of detail is unnecessary.

Are these five watersheds identified somewhere? Supplemental material?

How is foliar d15N on the same timescale as bulk soil d15N? The plant available portion of the soil pool is very small and is not what is measured by bulk soil. Foliar %N and d15N can vary on very short time scales. Bulk soil d15N may vary in response to disturbance, but the soil N pool is many orders of magnitude larger than the foliar N pool.

What does 'that' refer to in this sentence (that of water-based proxies)

Does the absence of a correlation between soil solution and stream DIN:DON suggest that stream DIN:DON does not reflect what is available in the terrestrial ecosystem?

Foliar 15N is not an integrator on the time scale of decades to centuries

It seems a fairly broad interpretation to say that these data suggest that correlations between categories 1 and 2 are robust—some of them may be, but not all of them. To what extent is it reasonable to extrapolate this finding?

I don't see why one would expect DIN:DON to be correlated with soil 15N, they are measuring very different things.

Is DIN:DON more sensitive to N deposition than DIN?

What does it tell you if soil solution DIN:DON is not correlated with stream DIN:DON?

Hydrologic flowpath and flowrate are also probably important.

2002-4 Work by K. Lohse et al. addresses these issues.

Figures and tables

Fig 2 there is a lot of useful information in Figure 2, but the graphs are too small and are illegible. The format, in the end, is more clever than useful. It would be better to enlarge the graphs a bit so that it is easier to resolve the patterns. (The quality of the figure in the paper I downloaded is fair, but I assume there is a high resolution version).

The size of the statistical info is fine and legible. It might be easier to follow if it were presented in the same triangle configuration as the figures (as opposed to flipped) or else in a table.

---

## Author Response (AR1)

**Author's Response to Comments by Referees**

We thank the reviewers for their thoughtful comments, which we think have led to substantial improvements in the manuscript. Below we provide a detailed list of responses.

**Anonymous Referee #1**

**1. Referee #1:** This study presents a comparison of different proxies for assessing the nitrogen status of terrestrial ecosystems. Such proxies are essential if we are to assess the influence of nitrogen on these ecosystems, but various proxies exist and it is yet unclear which proxies are to preferred above others, and how the different proxies relate to one another. The current study addresses primarily the latter. Although I find this a useful effort, I think especially the analyses require revisions to improve the quality of this study.

General comments: As the study is now it is only indicative of the relationship between different proxies of nitrogen availability, and although differences between biomes are discussed, I think additional analyses are required to be able to draw firm conclusions regarding the difference among e.g. boreal, temperate and tropical systems. So, I suggest that the authors test statistically if for example the relationship between soil d15N soil and foliar d15N differs between temperate and tropical systems (as suggested on l. 167-168).

**Response:** We thank you for this suggestion and agree with the idea. Although we lack statistical power in boreal regions to test how these relationships change, we do discuss how they differ between temperate and tropical ecosystems in lines 177-183. We have also included the following text to address the way in which these relationships change:

Lines 179-185: "Some relationships between proxies differed with latitude. Soil and foliar $\delta^{15}N$ were more tightly correlated in the tropics (n=24, $\tau$=0.68, $p$<0.0001) than in the temperate zone (n=49, $\tau$=0.23, $p$=0.02). Soil $\delta^{15}N$ was correlated with net nitrification in tropical (n=17, $\tau$=0.39, $p$=0.03), but not temperate regions. Conversely, soil $\delta^{15}N$ was correlated with net N mineralization (n=44, $\tau$=0.34, $p$=0.001) in temperate but not tropical areas. Stream DIN:DON was correlated with net nitrification (n=10, $\tau$=0.63, $p$=0.01) and N mineralization (n=10, $\tau$=0.78, $p$=0.002) in the temperate zone, and not in the tropics (n=4, $p$>0.05)."

Lines 241-250: "While most observed correlations were consistent across latitudes, a few differed between the tropics and the temperate zone. The correlations of soil $\delta^{15}N$ with foliar $\delta^{15}N$, foliar $\delta^{15}N$ with net nitrification, and net nitrification with N mineralization were consistent across both tropical and temperate regions. Net nitrification and N mineralization were correlated with stream DIN:DON only in temperate regions. These data suggest that while terrestrial proxies may be a useful across biomes, stream DIN:DON requires further research to understand the extent of its applicability across space. The correlation between foliar and soil $\delta^{15}N$ also differs across latitudes, in that the correlation in the tropics was much tighter than in the temperate zone. Bias in the literature towards natural abundance isotopic data from the temperate zone may explain why previous research looking at this relationship has been noisy (Craine et al., 2009)."

Finally, we note that we did not test these relationships using regression, rather non-parametric correlation. Other than comparing the strength of the correlation, this makes it difficult to determine statistically whether the nature of the relationships vary with biome.

**2. Referee #1:** Although I understand that the potential proxies are numerous and choices need to be made, I wonder why authors did not include soil C:N ratio. This ratio is often considered a good indicator of soil nutrient status (see e.g. Alberti et al 2015, iForest 8, 195-206). It is also an easy measurement to make so I advocate to include this ratio in the study.

**Response:** We agree, soil C:N would be another useful proxy, since as the reviewer notes it is commonly measured, and is highly correlated with soil and plant $\delta^{15}N$ (Craine et al. 2015, Plant and Soil, (369): 1-26). However, at this stage including C:N would require us to withdraw the manuscript and revisit the literature, an exercise that might prove useful but our suspicion is would not change the conclusion of this manuscript. We have however updated the sentence in lines 90-94 to say: "We chose these metrics because 1) other authors have suggested that they are indicative of soil nutrient status (Martinelli et al., 1999, Amundson et al., 2001, Brookshire et al., 2012; Figure 1), and 2) they are thought to integrate N fluxes on different timescales (e.g. soil $\delta^{15}N$ integrates N losses over decades while net N mineralization rates integrate inorganic N production over days; Binkley and Hart, 1989, Hogburg 1997)."

**3. Referee #1:** I wonder if the authors could test for the influence of the distance between watershed and stream in the analysis of the correlation between soil proxies versus stream DIN:DON (perhaps with a subset for which this info is available).

**Response:** Unfortunately distance from the stream is rarely reported, especially when terrestrial metrics are gathered from different papers than that of water-based metrics (which was typical). We have updated the text to include this point in lines 123-124: "Terrestrial metrics were typically gathered from different papers than that of water-based metrics, requiring validation of congruent watershed location."

Another issue along the same lines it that other land-uses in the greater watershed are rarely reported either, which can potentially influence the relationship between soil and stream metrics. We revised the text to raise this point in lines 238-240: "varied land-use (e.g. pasture, N fixing plant species, etc.) upstream of undisturbed sites is typically not reported in the literature, but is another possible explanation for the break down between terrestrial and water-based proxies."

**4. Referee #1:** Finally, I suggest that the authors make their dataset available. At minimum, a list of sites and coordinates should be given, but I hope authors can also provide the associated data for the various proxies.

**Response:** Along with the manuscript, we submitted a supplementary zip file with the complete data set, including a list of sites and complete citations for papers we extracted data from. We apologize if this was not made available and will inquire with the editor about its status. We also plan to include an additional figure that outlines the approximate location of our sites (see Fig. 2b).

Figure 2. a) Distribution of grassland (grey) and forest (black) watershed mean annual temperature (MAT; °C) and mean annual precipitation (MAP; mm yr$^{-1}$) included in meta-analysis (left), and b) location of 154 sites (some black dots represent multiple watersheds; right).

[Figure]

[Figure]

**5. Referee #1:** Specific comments:
l. 9: bracket after 'correlate' should come before 'correlate'
l. 46: remove 'in solution' (dissolved is obviously in solution)
l. 70: I suggest to rephrase 'we asked which were correlated'

**Response:** We corrected these. Thank you.

**6. Referee #1:**
l.181-182: I disagree with this statement that category 1 and 2 proxies showed a robust relationship. In a dataset with huge variation (like this one), a significant correlation between 2 metrics does not necessarily imply a robust relationship, and visual inspection of the relationships in Fig. 3 does not support the statement. There is some correlation, but there is a lot of unexplained variance.

**Response:** We agree this statement was misleading as stated. We have adjusted the text to point out the statistical significance and removed the word "robust". The sentence (lines 191-192) now reads: "Our data suggest that category 1 and 2 metrics are correlated".

**7. Referee #1:** l.194- 198: How is d15N of soil (and plants) influenced by N deposition? If N deposition influences soil and plant d15N, the use of these proxies globally may not be ok because similarly rich sites would differ in d15N depending on the N source.

**Response:** This is a good point. Our data show no evidence of N deposition on plant or soil $\delta^{15}$N, but many of our sites do not have N deposition data. We have revised the text to reflect this in lines 206-209: "In our dataset, N deposition was not correlated with stream DIN:DON ($\tau$=0.03, $p$>0.05), or any other metric. Thus our data do not support the idea that N deposition is responsible for the lack of correlation between these two long-term proxies."

**8. Referee #1:**
l.211: 'that' after 'seem' and 'to' after 'used

**Response:** We corrected this. Thank you.

**Anonymous Referee #2**

**1. Referee #2:** BG review This paper poses an interesting and important question about whether different metrics used to characterize N availability (which represent different spatial and temporal scales) are correlated. This topic is of potential interest to a broad group of researchers who consider N availability in their studies. The paper attempts to evaluate some underlying assumptions that are included implicitly or explicitly in interpreting ecosystem N dynamics. The scope of the analysis is not really clear from the paper. It is a little surprising that the authors did not include several large syntheses of similar data (Aber at al. 2003, BioScience, Pardo et al. 2006 Biogeochemistry, the CANIF study in Europe, Schulze 2000 Springer).

**Response:** Thank you for this comment and for the citations. We did our best to find all the literature available, however we are sure to have missed some. We did not include Aber et al. (2003) because they do not report the variables we are focused on, at least not in a manner that was useable in this analysis (i.e. they report $NO_3^-$ but not $NH_4^+$ or DON, and they report percent nitrification rather than a nitrification rate in ug N/g/d). We did use Pardo et al. 2006 (which included CANIF sites) to find original papers, from which we extracted data. Any sites that seem to be excluded may have been done because we were unable to find multiple proxies from that site. We thank you for the Schulze 2000 reference and have added these data to our analysis.

**2. Referee #2:** The scope of the analysis is important, because it can be difficult to make assertions about different climatic zones or life forms unless enough variation is included among the samples to represent that observed

**Response:** We agree and have included the following text to strengthen that point.

Line 76-79: "This review assesses the correlation between common foliar, surface soil (i.e. $\delta^{15}N$, nitrification and mineralization), and nutrient loss (i.e. soil solution and stream N concentrations) metrics of N availability from unmanaged ecosystems globally."

**3. Referee #2:** Several issues should be addressed: Nitrate leaching is referred to as if it were the driver of the fractionation that would lead to 15N-enrichment of material remaining in the ecosystem (soil, foliage). In fact the elevated nitrification which leads to an increase d15N of the plant available (including the nitrate that leaches to the stream) is the driver. The authors are, no doubt, well aware of this, but it is worth taking the trouble to be more precise for the reader less familiar with these dynamics. This should be addressed at several points in the paper.

**Response:** Thank you. We did not intend to give the impression that leaching is a key driver of isotopic enrichment in leaves and soil. To remedy this, we inserted: "primarily denitrification" in lines 61-62, and "during nitrification" in line 65.

**4. Referee #2:** The isotope literatures is not as current as it could be. I have given some examples of possible additional citations.

**Response:** Thank you for those. We have included the caveat that our literature search included only papers published prior to 2013 (line 84).

**5. Referee #2:** I assume that when the authors talk about long-term patterns and measures that are invariant temporally, that they mean in undisturbed systems. This should be stated explicitly, since over the long term, at many of these study sites, various disturbances have occurred which disrupt that N cycle and which would affect the values of these metrics.

**Response:** We added "in relatively undisturbed ecosystems" (line 32). However, one weakness of our approach for stream measurements is the nature of land use change upstream from a particular site described in the papers we searched. We have made that caveat clearer on lines 238-240, which now reads: "varied land-use (e.g. pasture, N fixing plant species, etc.) upstream of undisturbed sites is typically not reported in the literature, but is another possible explanation for the lack of correlation between terrestrial and water-based proxies."

**6. Referee #2:** Need to define what is meant by N status.

**Response:** Agreed. We rephrased this to include "relative abundance of plant available N" (line 31-32).

**7. Referee #2:** More explanation about the differences between observed correlations in tropical versus temperate systems would be useful (why were foliar and soil d15N correlated in tropical, but not temperate?)

**Response:** We added lines 241-250: "While most observed correlations were consistent across latitudes, a few differed between the tropics and the temperate zone. The correlations of soil $\delta^{15}N$ with foliar $\delta^{15}N$, foliar $\delta^{15}N$ with net nitrification, and net nitrification with N mineralization were consistent across both tropical and temperate regions. Net nitrification and N mineralization were correlated with stream DIN:DON only in temperate regions. These data suggest that while terrestrial proxies may be a useful across biomes, stream DIN:DON requires further research to understand the extent of its applicability across space. The correlation between foliar and soil $\delta^{15}N$ also differs across latitudes, in that the correlation in the tropics was much tighter than in the temperate zone. Bias in the literature towards natural abundance isotopic data from the temperate zone may explain why previous research looking at this relationship has been noisy (Craine et al., 2009)."

**8. Referee #2:** Abstract:
10 if space permit, include the region considered in this study
19 is there a 'that' missing? i.e., given that both. . .

**Response:** We corrected these. Thank you.

**9. Referee #2:** 27 why 'Nevertheless'? what follows doesn't not seem to contrast with what was said in the first sentence.
31 don't really need 'such' on this line

32 I would suggest adding 'rates' after mineralization and nitrification, to make the comparison to another flux clearer. Also, the verb needs to agree with the subject is→are

**Response:** We corrected these. Thank you.

**10. Referee #2:** 33-4 This is an important point (basing annual budgets on short-term measures) and one that is often ignored.

**Response:** Agreed.

**11. Referee #2:** 34-6 There seems to be a word missing or a punctuation problem. Is 'are relevant' associated with scales or N status?

**Response:** We changed this to "While N status measured over longer temporal and larger spatial scales is relevant to many ecosystem properties and their response to global change, it is more difficult to measure." (lines 35-37)

**12. Referee #2:** 87-90 Is this level of detail necessary?

**Response:** See our response to Reviewer 1 (and our response now in lines 90-94), who had a query about why we chose these metrics and not others. We thought it best to present our thinking as fully as possible.

**13. Referee #2:** 90 What is meant by 'intact'? does this mean 'not fragmented'? Or is it intended to include disturbance as well? And if so, only anthropogenic disturbance (e.g., harvesting) or also natural (fire, wind, ice or pest events, etc.)?

**Response:** We agree "intact" was unclear. We changed this to: "We limited our search criteria to studies that took place in forest or grassland ecosystems that had not incurred any large disturbances that might impair their function." (lines 98-100).

**14. Referee #2:** 92 Is there a list of the sites in supplemental information? (cite supplemental material here)

**Response:** The list is available in the supplemental. We have now noted this on line 102. Thank you.

**15. Referee #2:** 101 Is it appropriate to lump net nitrification potential measures with measures of nitrification? This should be justified.

**Response:** We agree that this is a point worth clarifying in the text. We limited our nitrification methods to intact soil core, buried bag, and lab incubations in order to avoid any methodological differences (as state in lines 110-112). In the literature net nitrification and nitrification potential are terms that are sometimes (but not always) used interchangeably (Ross et al. 2012, Journal of Geophysical Research: Biogeosciences, 117(G1); Bohlen et al. 2001, Ecology, 82(4), 965-978). Some authors define a buried bag incubation as "potential" because it is not what is actually

happening in intact soils. However, others define nitrification "potential" as how much nitrification happens when soils are amended to overcome potential substrate limitations to nitrification. We did not include any nitrification assays where the soils were amended and have thus revised the text to reflect this (lines 110-112): "In order to control for methodological differences, we limited our net nitrification and N mineralization methods to those which used intact soil core, buried bag, and laboratory incubations of unamended soils".

**16. Referee #2:** 105-6 This level of detail is unnecessary.

**Response:** We thought that including this level of detail might help field reader questions regarding the analyses we chose to run and have chosen to leave the text as is unless the editor prefers we remove it.

**17. Referee #2:** 107 Are these five watersheds identified somewhere? Supplemental material?

**Response:** Yes, the supplemental data lists full citations for each watershed, and there we state where we "collected soil".

**18. Referee #2:** 137 How is foliar d15N on the same timescale as bulk soil d15N? The plant available portion of the soil pool is very small and is not what is measured by bulk soil. Foliar %N and d15N can vary on very short time scales. Bulk soil d15N may vary in response to disturbance, but the soil N pool is many orders of magnitude larger than the foliar N pool.

**Response:** We agree that foliar $\delta^{15}N$ can differ among species, and that N in leaves turns over much more quickly than N in soil. However, our understanding is that average foliar $\delta^{15}N$ for a site is relatively stable in time, absent large changes in species composition. If the reviewer can point us to literature that suggests otherwise we would be happy to incorporate it in our discussion.

**19. Referee #2:** 141 What does 'that' refer to in this sentence (that of water-based proxies)

**Response:** We changed this to read "with water-based proxies" (line 166).

**20. Referee #2:** 144 Does the absence of a correlation between soil solution and stream DIN:DON suggest that stream DIN:DON does not reflect what is available in the terrestrial ecosystem?

**Response**: We think so, at least for the dataset here. We discuss this in lines 210-225, which read: "Another surprise from our dataset is that soil solution DIN:DON was not significantly correlated with any other metric, not even with stream DIN:DON, despite ~40% of papers in our dataset reporting both soil solution and stream DIN:DON in the same watershed. While the correlation between soil solution DIN:DON below the rooting zone and N availability has been documented across gradients in soil age and fertility (Hedin et al., 1995), this correlation was not found across the range of sites examined here. We found no relationship between soil solution DIN:DON and lysimeter depth, suggesting that the majority of N transformations responsible for the discontinuity between soil solution DIN:DON and that of terrestrial metrics are likely

occurring either within the rooting zone or in riparian zones. Neither soil solution or stream DIN:DON was sensitive to environmental variability (i.e. elevation, temperature, precipitation, N deposition), suggesting that processing along flow paths may be responsible for the disconnect between soil solution and stream N concentrations. From these data, at least, it does not seem that soil solution DIN:DON can be used to infer terrestrial N status across this suite of unmanaged sites. These data also do not support the idea that soil solution DIN:DON is representative of N forms that leach into streams (Binkley et al., 1992; Pregitzer et al., 2004; Fang et al., 2008)."

**21. Referee #2:** 155 Foliar 15N is not an integrator on the time scale of decades to centuries

**Response:** The fact that foliar N is derived from soil N and that foliar $\delta^{15}N$ correlates with soil $\delta^{15}N$ across broad spatial scales suggests that these two values are dependent on one another. While we agree that average foliar $\delta^{15}N$ may change faster than soil $\delta^{15}N$ in perturbed sites that have experienced a turnover in species composition or large scale disturbances, we argue that foliar $\delta^{15}N$ in relatively undisturbed ecosystems (such as the sites that we analyzed) change on a similar timescale as soil $\delta^{15}N$.

**22. Referee #2:** 160 It seems a fairly broad interpretation to say that these data suggest that correlations between categories 1 and 2 are robust some of them may be, but not all of them. To what extent is it reasonable to extrapolate this finding?

**Response:** We agree that it was not well written. We have changed this line to read "Our data suggest that category 1 and 2 metrics are correlated" (lines 191-192).

Since these data incorporate as much of the available data that we could find across a broad geographic and climatic range, we would imagine these findings can be extrapolated, but when we look at differences within biomes it becomes apparent that these relationships may vary geographically, and for that reason we call for more research examining these relationships at smaller spatial scales in lines 254-257: "Explicit comparisons of these proxies to each other, with a focus on how they are influenced by hot-spots, hot-moments, biological diversity, and N transformation between the soil-stream interface, will enhance their utility for understanding N availability at the ecosystem scale."

**23. Referee #2:** 171 I don't see why one would expect DIN:DON to be correlated with soil 15N, they are measuring very different things.

**Response:** We agree. However, in the literature both are used as an indication of N status within watersheds. We hope that these data highlight that they are measuring different things, and that interpretations of terrestrial N status based on these metrics is not straightforward. This is a key point we hope this paper makes.

**24. Referee #2:** 178 Is DIN:DON more sensitive to N deposition than DIN?

**Response:** We would presume that DIN is more sensitive than DIN:DON, because N can be deposited in both forms, but the majority is deposited as DIN. As we suggest in lines 205-206,

because most N deposition comes in the form of DIN, DIN:DON is lower in pristine settings than in polluted ones.

Lines 205-206: "We note that stream DIN:DON is sensitive to N deposition, and that relatively pristine settings have a lower DIN:DON than polluted ones (Perakis and Hedin, 2001)."

**25. Referee #2:** 183 What does it tell you if soil solution DIN:DON is not correlated with stream DIN:DON?

**Response:** We propose several explanations for this in the text (lines 226-240). One of which is that N is removed along hydrologic flow paths, and another is that stream N is potentially affected by upstream land-use/inputs that overshadow local inputs.

**26. Referee #2:** 198 Hydrologic flowpath and flowrate are also probably important. 2002-4 Work by K. Lohse et al. addresses these issues.

**Response:** Agreed. Thank you for the citation. We touch on this in lines 226-240: "While nitrate ($NO_3^-$) removal along flow paths can reduce stream $NO_3^-$ (Vidon et al., 2010), with higher percent removal in forested watersheds (Sudduth et al., 2013), DON has been shown to be relatively resistant to removal by decomposition and biologic uptake along subsurface flow paths (Carreiro et al., 2000, Neff et al. 2003). We found no correlation between stream and soil solution DIN:DON, and suggest that variation in $NO_3^-$ removal (relative to DON) along flow paths of undisturbed ecosystems may explain this lack of correlation. The extent to which riparian zones influence nutrients varies spatially with geomorphology, soil texture, vegetation, and riparian zone development (McDowell et al., 1992, Mayer et al., 2007); and soils with high rates of leaching to ground water may bypass riparian processing. As nutrients leach down the soil profile, denitrification, biologic uptake, and storage are all potential mechanisms that could alter soil solution and stream N species concentrations. Investigation of soil profile processes and riparian zone spatial variability may help determine where and when watershed-scale N status can be inferred from these proxies. Alternatively, varied land-use (e.g. pasture, N fixing plant species, etc.) upstream of undisturbed sites is typically not reported in the literature, but is another possible explanation for the break down between terrestrial and water-based proxies."

**27. Referee #2:** Figures and tables
Fig 2 there is a lot of useful information in Figure 2, but the graphs are too small and are illegible. The format, in the end, is more clever than useful. It would be better to enlarge the graphs a bit so that it is easier to resolve the patterns. (The quality of the figure in the paper I downloaded is fair, but I assume there is a high resolution version).
The size of the statistical info is fine and legible. It might be easier to follow if it were presented in the same triangle configuration as the figures (as opposed to flipped) or else in a table.

**Response:** Thanks. Since there are 16 graphs, we chose this format to conserve space. However, we have attempted to make the panels bigger so that they can be seen more easily. We hope that the editor will inform us of any further issues with legibility/resolution.

**Measuring ecosystem nitrogen status: a comparison of proxies**

Maya Almaraz[1], Stephen Porder[1]

[1]Department of Ecology and Evolutionary Biology, Brown University, Providence, 02912, USA

*Correspondence to:* Maya Almaraz (maya_almaraz@brown.edu)

**Keywords:** nitrogen availability, nutrient limitation, $\delta^{15}N$, nitrogen mineralization, dissolved organic nitrogen

**Abstract**. There are many proxies used to measure nitrogen (N) availability in watersheds, but the degree to which they do (or do not) correlate within a watershed has not been systematically addressed. We surveyed the literature for intact forest or grassland watersheds globally, in which several metrics of nitrogen availability have been measured. Our metrics included: foliar $\delta^{15}N$, soil $\delta^{15}N$, net nitrification, net N mineralization, and the ratio of dissolved inorganic to organic nitrogen (DIN:DON) in soil solution and streams. Not surprisingly, the strongest correlation (Kendall's tau) was between net nitrification and N mineralization ($\tau=0.71$, $p<0.0001$). Net nitrification and N mineralization were each correlated with foliar and soil $\delta^{15}N$ ($p<0.05$). Foliar and soil $\delta^{15}N$ were more tightly correlated across tropical sites ($\tau=0.68$, $p<0.0001$), than in temperate sites ($\tau=0.23$, $p=0.02$). To our surprise, the only significant correlations we found between terrestrial- and water-based metrics were that of net nitrification ($\tau=0.48$, $p=0.01$) and N mineralization ($\tau=0.69$, $p=0.0001$) with stream DIN:DON. The relationship between stream DIN:DON with both net nitrification and N mineralization was significant only in temperate, but not tropical regions. Given that both soil $\delta^{15}N$ and stream DIN:DON are used to infer long-term N status, their lack of correlation in watersheds merits further investigation.

**Commented [AM1]:** Revised text is signified by reviewer number (R1 or R2) and comment number (as they appear in the "Response to Reviewers" document)

**Commented [AM2]:** R1.5

**Commented [AM3]:** R2.8

**Commented [AM4]:** R2.8

**1.0 Introduction**

Nitrogen (N) limitation to primary production is widespread in both terrestrial and aquatic ecosystems, and variation in N availability drives differences in ecosystem properties across space and time (Vitousek and Howarth, 1991; Elser et al., 2007; LeBauer and Treseder, 2008). Yet quantifying N availability over timescales that are relevant in ecosystems is non-trivial. Short timescale measurements of N availability in soil are common (e.g. inorganic N pools, N mineralization and nitrification rates; Binkley and Hart, 1989; Sparks et al., 1996), but such short-term proxies are influenced by both short and long-term drivers, and thus it is difficult to know whether short-term proxies can be used to infer N status (i.e. the relative abundance of plant available N) over long timescales in relatively undisturbed ecosystems. For example, measured net mineralization and nitrification rates in arctic tundra are commonly less than annual plant uptake (Schimel et al., 1996; Schmidt et al., 1999), and annual N budgets based on short-term measurements are difficult to balance (e.g. Magill et al., 1997). While N status measured over longer temporal and larger spatial scales is relevant to many ecosystem properties and their response to global change, it is more difficult to measure.

Land-based investigations of N cycling commonly measure extractable N, N mineralization, and nitrification, which give a snapshot of N status over minutes to days (Binkley and Hart, 1989; Robertson et al., 1999). Some researchers also use lysimeters to quantify dissolved N losses from below the rooting zone (Hedin et al., 2003; McDowell et al., 2004; Lohse and Matson, 2005) on a similar timescale. Repeated measurements give longer timescale information, but even the longest studies are short relative to ecosystem development.

In addition to these short-term proxies, there are two relatively common measurements that are thought to average over space and/or time. The first is the ratio of dissolved inorganic

*Commented [AM5]: R2.9*

*Commented [AM6]: R2.9*

*Commented [AM7]: R2.6*

*Commented [AM8]: R2.5*

*Commented [AM9]: R2.9*

*Commented [AM10]: R2.11*

[revised manuscript text omitted]

Commented [AM36]: R1.4, R2.1, R2.17

---

## Author Response (AR2)

[revised manuscript text omitted]

**Reply to the Associate Editor's Comments**
Thanks very much for your continued handling of the manuscript and your constructive
comments. Please find attached our responses point by point. We hope you find them
satisfactory.
**1) Editor:** While I can see that you do not want to invest into gathering C:N ratio, I agree with
reviewer #1 that this would have improved the impact of your manuscript substantially. This is
of course your decision. However, either in the introduction or discussion you need to explain
why you have omitted one of this easily measurable and relevant measure of nitrogen status, and
why you think that this would not have affected your results.
**Response:** We agree that the inclusion of C:N data would have been an excellent addition to the
manuscript, and we regret not having included it in the original analysis. Unfortunately, to redo
the analysis using our original method would require performing a new search for every paper
that reports C:N, following up on new watersheds included in the data set to find additional
metrics measured there, and ultimately retracting the current version of the manuscript from
Biogeosciences, as we estimate this amount of work taking months, if not longer. A meta-
analysis of C:N has never been done and could potentially hold up as a paper on its own. Soil
C:N is not usually used as a long term proxy for N availability in the same way as $\delta^{15}$N, because
C:N is not solely sensitive to fractionating N losses. Further research looking at soil C:N
dynamics at the global scale will shed light on the potential utility of this metric in assessing long
term N availability. We want to acknowledge the importance of this metric in our discussion and
have thus added the following paragraph to our discussion section:
Lines 262-277: "One commonly reported metric that was not included in our analysis is the bulk
soil carbon to nitrogen ratio (C:N). The conception for this manuscript focused on the
relationship between soil $\delta^{15}$N and stream DIN:DON, because these are most commonly used as
long term proxies of N availability (Martinelli et al., 1999; Amundson et al., 2000; Perakis and
Hedin et al. 2001; Brookshire et al. 2012). Specifically, theory regarding spatial differences in N
availability, especially between the tropics and temperate zone, focus on the metrics we report
here. Conclusions about N richness at the global scale have yet to use C:N data to support the
theory for latitudinal gradation in N availability (Brookshire et al. 2011; Smith et al.; 2014). Soil
C:N has already been shown to be tightly correlated with soil $\delta^{15}$N at the global scale (Craine et
al., 2015), but has yet to be compared to the other metrics we present here. It's relationship with
soil $\delta^{15}$N leads us to believe that soil C:N will likely reflect the same trends as that of soil $\delta^{15}$N.
The measurement of soil C:N is perhaps reported more so than any other biogeochemical metric,
and certainly more so than those included in this meta-analysis. We suggest that future research
utilize meta-analysis techniques to look at how soil C:N changes across ecosystem gradients, and
whether or not it agrees with latitudinal patterns observed for soil $\delta^{15}$N and stream DIN:DON
(Martinelli et al., 1999; Brookshire et al. 2011).
**2) Editor:** L177ff: Please acknowledge that lacking data availability prevented you from
identifying differences between temperate and boreal ecosystems (rather than not being there at
all).

**Response:** We have added the following text to our results in lines 190-192: "Because we only
found multiple proxies measured at eleven boreal sites, this limited our ability to compare
correlated data in boreal regions with correlations in temperate or tropical areas."
**3) Editor:** L205ff: This discussion fails to acknowledge that N deposition data was lacking for
an important fraction of the data set.
**Response:** This is a good point. We have added the following text to the discussion in lines 217-
219: "Although 48% of our sites lacked N deposition data, our data do not support the idea that
N deposition is responsible for the lack of correlation between these two long-term proxies."
**4) Editor:** Figure 2: increase font size of axis labels
**Response:** Change made.
**5) Editor:** Figure 3: Improve resolution of figure. Use entire column width to maximise
readability.
**Response:** Change made. All of our figures are now 600dpi, please let us know if that resolution
does not suffice.

---

## Author Response (AR3)

[revised manuscript text omitted]

**Reply to the Associate Editor's Comments**
**Editor:** p.13, line 271, It's → Its
**Response**: We have changed this in the text.  Thank you for catching this error.